# Estimating the Impact of Vaccination Campaigns on Measles Transmission in Somalia

**DOI:** 10.3390/vaccines12030314

**Published:** 2024-03-16

**Authors:** Niket Thakkar, Ali Haji Adam Abubakar, Mukhtar Shube, Mustafe Awil Jama, Mohamed Derow, Philipp Lambach, Hossam Ashmony, Muhammad Farid, So Yoon Sim, Patrick O’Connor, Anna Minta, Anindya Sekhar Bose, Patience Musanhu, Quamrul Hasan, Naor Bar-Zeev, Sk Md Mamunur Rahman Malik

**Affiliations:** 1Institute for Disease Modeling, Bill & Melinda Gates Foundation, Seattle, WA 98109, USA; 2Federal Ministry of Health, Mogadishu P.O. Box 22, Somalia; 3World Health Organization, 1202 Geneva, Switzerland; 4World Health Organization, Mogadishu 63565, Somalia; 5Gavi, The Vaccine Alliance, 1218 Geneva, Switzerland; 6World Health Organization, Regional Office for the Eastern Mediterranean, Cairo 11371, Egypt

**Keywords:** measles, vaccination campaign, transmission model, serology, Somalia

## Abstract

Somalia is a complex and fragile setting with a demonstrated potential for disruptive, high-burden measles outbreaks. In response, since 2018, Somalian authorities have partnered with UNICEF and the WHO to implement measles vaccination campaigns across the country. In this paper, we create a Somalia-specific model of measles transmission based on a comprehensive epidemiological dataset including case-based surveillance, vaccine registries, and serological surveys. We use this model to assess the impact of these campaign interventions on Somalian’s measles susceptibility, showing, for example, that across the roughly 10 million doses delivered, 1 of every 5 immunized a susceptible child. Finally, we use the model to explore a counter-factual epidemiology without the 2019–2020 campaigns, and we estimate that those interventions prevented over 10,000 deaths.

## 1. Introduction

Measles is a highly contagious airborne disease that can lead to severe complications and death. In 2023 more than 240,000 suspected measles cases were reported globally by World Health Organization (WHO) member states, a 40% increase from the 170,000 reported in 2022. Each of these cases of illness represents a significant threat to health and wellbeing, and all of those attributable to measles are preventable with two doses of a safe and effective vaccine [1].

The vast majority of measles complications and deaths occur in countries with fragile health infrastructures that struggle to routinely vaccinate children [2]. Of the children left susceptible, 75% are infected before turning five [3]. Upon infection, measles suppresses an individual’s immune system, and the likelihood of severe outcomes is elevated by malnutrition and other systemic health issues [4]. As a result, in the world’s most challenging settings, risks compound and effective vaccine delivery is a high-priority global health and human rights challenge.

Somalia is an example of one such fragile setting. On a backdrop of protracted armed conflict, civil war, and political instability since the 1990s, Somalia has more recently experienced several years of climatic shocks such as droughts and floods. In 2022, the country was hit by one of the most severe droughts in its history, and Somalia’s Food Security and Nutrition Analysis Unit estimated that 54.5% of Somali children under five suffered from acute malnutrition [5]. Meanwhile, measles has been endemic in Somalia for years and over 23,000 suspected measles cases were reported in 2017 [6], demonstrating the country’s potential for sweeping and disruptive outbreaks. With additional COVID-19 related disruptions to routine immunization services since then [7], the population’s risk of measles outbreaks and severe outcomes has been critical since 2020.

In response to these issues, Somalia’s ministries of health partnered with UNICEF and the WHO to conduct measles vaccination campaigns, integrated with other health services like vitamin A supplementation, across the country in 2018, 2019–2020, and in 2022. Together, these campaigns delivered more than 10 million measles vaccines to children under 10. This was an ambitious immunization effort in a uniquely challenging setting, complicated further by the need for COVID-19 safety precautions. Documenting and understanding the campaigns’ implementation details and epidemiological implications represents an opportunity for us to gain insight into vaccine delivery in fragile settings.

To that end, in this Communication, we use modern disease modeling methods to assess the impact of these campaigns on measles transmission in Somalia. Briefly, taking a comprehensive epidemiological dataset as input, including data from case-based surveillance, vaccine registries, and serological surveys, we construct a Somalia-specific model of measles transmission from 2018 to 2023. We use this model to estimate the impact campaign doses had on measles susceptibility, and we compare the current situation to a model-based counter-factual epidemiology without the campaigns. This assessment illustrates that Somalia’s vaccination efforts have prevented thousands of measles infections from 2019 to 2023.

## 2. Measles Epidemiology and Immunization in Somalia

Key components of the epidemiological dataset are visualized in Figure 1. In the top panel, suspected measles cases from 2018 to 2023 (red) illustrate the decline in cases after the 2017 outbreak, followed by a resurgence in 2022. The majority of cases were in children younger than five (pink), with a median age at infection of 2.4 years.

In the second panel, the administrative estimates of yearly live births and surviving infants are plotted on the same semi-monthly increments as cases, under the assumption that the seasonality in birth rates is negligible. We see that birth cohorts have grown in size since 2020, in turn increasing the size of the population at risk of measles infection.

Figure 1’s third panel summarizes Somalia’s immunization efforts. In purple and yellow are the routine first (measles-containing-vaccine, MCV1) and second (MCV2) doses administered and reported by health facilities, showing that MCV2 was partially introduced in late 2021. Meanwhile, campaign efforts are marked on the same timeline by black bars sized in proportion to the number of doses delivered.

Both the 2018 and 2022 campaigns had a nationwide scope, with the 2018 campaign targeting children up to age 10 and the 2022 campaign up to 5. Meanwhile, in contrast, partly for security reasons, the efforts in 2019–2020, also targeting children under 5, were implemented in geographically distinct phases. All southern states except Banadir implemented the campaign in November 2019; Puntland and Somaliland implemented in the last week of March 2020, and Banadir implemented in August and September 2020.

We incorporate some additional data into the transmission model as well, specifically an estimate of the population pyramid [8] and the outcomes of a 2021 serological survey. The latter are visualized in Figure 2 (black), illustrating that nearly 95% of 15–29-year-olds had measles immunity either through vaccination or a prior infection [9].

To disentangle these distinct contributions, we assume that, conditional upon being unvaccinated, the age-at-infection is log-normally distributed. The resulting model, with an immunity component taken from the MCV1 data above, under the assumption that MCV1 seroconverts 85% of the time, is illustrated in beige. The binned fit (red dashed line) captures the serological data and gives an inferred estimate of the age-at-infection distribution.

We can validate this inference through a comparison with the observed age distribution in the surveillance data. This is shown in Figure 2’s second panel, where the age-at-infection distribution (beige) is binned (red) to predict the age distribution of cases (grey). The first two bins, capturing 99% of cases, are accurately predicted by the model. The discordance in subsequent bins suggests that case-based reporting under-represents measles infections in older age groups, in qualitative agreement with the findings from other settings [10]. Taken as a whole, this exercise shows that the immunization and serological data can be reconciled into a consistent epidemiology.

## 3. Transmission Modeling

The transmission model offers a mathematical platform to reconcile the data more broadly across Figure 1 and Figure 2 into a single stochastic process. At a high level, following [11,12], we assume that individuals can be meaningfully compartmentalized into susceptible, infectious, and recovered states on a semi-monthly timescale [13,14]. Surviving births are either immunized via routine vaccination or else become susceptible and remain at risk of getting measles through interactions with infectious individuals. New infections transition to being infectious within a semi-month before being removed from the system. Finally, during the five measles campaigns indicated in Figure 1, susceptible individuals have the opportunity to be immunized.

The model is fit to Somalia’s case data using the approach from Ref. [12], adapted to include vaccination along the lines of Ref. [11] and leveraging the age-at-infection distribution from Figure 2. We assume that the under-reporting of infections happens at a constant but unknown rate on average over the 5-year model period, and we further assume that the fraction of susceptible–infectious pairs leading to new infections is log-normally distributed with a seasonally varying mean. Campaigns are defined in the model by an efficacy parameter which represents the fraction of delivered doses that immunized susceptible individuals.

The model’s fit and related inferences are visualized in Figure 3. In the top panel, with a time-averaged reporting rate of 5.3% (95% confidence interval (CI) 4.6% to 6.3%), infections in the model (red, 95% CI shaded) follow the variation in observed cases (black dots) closely. Driving that process is the underlying susceptibility of the population, visualized as a percentage of the total population in the second panel, clearly demonstrating persistent growth due to unvaccinated births and isolated declines due to the vaccination campaigns. The susceptibility estimate agrees well with the population-averaged serological survey results (black line, 95% CI shaded) despite being fit to only the case data directly. Taken as a whole, the model represents a plausible transmission process consistent with the full epidemiological dataset from Somalia.

Estimates of the fraction of campaign doses that immunized a child are visualized in Figure 3’s lower panels. Doses delivered in the 2018 campaign had little effect on immunity, as we might have expected in the wake of the large 2017 outbreak and with a much larger age range. In contrast, the 2019–2020 campaigns immunized large cohorts, and their geographic phases have distinct and intuitive behaviors. Specifically, in the southern states and Banadir, where routine vaccination coverage is lower, campaign doses had a higher likelihood of reaching a susceptible child. Meanwhile, in 2022, with much lower infection-derived immunity, the nationwide campaign was significantly more effective per dose than in 2018.

Taken together, from 2018 onwards, we estimate that on average one of every five campaign vaccines immunized a susceptible child. Under conservative assumptions of the measles infection–fatality ratio [15] and ignoring the substantial morbidity in children who survive measles, this implies that Somalia’s campaigns have averted 1 disability-adjusted life year (DALY) for every nine doses delivered, making them an exceptionally cost-effective intervention [16].

## 4. From Campaign Immunity to Averted Burden

The fitted model gives us a platform from which to estimate the transmission process in the absence of a campaign, all else being the same [17]. We apply this idea to the 2019–2020 campaigns in this section.

The original model (grey) and observed data (black dots) are visualized alongside two alternative scenarios in Figure 4. The expected trajectory in the absence of the campaigns (yellow) shows that Somalia was at risk of a 2022-like outbreak in 2020. This estimate is driven by the susceptibility estimate from Figure 3, which suggests that susceptibility was above 2022 levels in late 2019 and that the 2019–2020 campaigns were timely. Meanwhile, a scenario with 50% more effective campaigns (green) shows that agreement with the case data constrains efficacy—the observed 2022 outbreak limits the 2019–2020 campaigns’ effect on susceptibility.

Comparing model trajectories with and without the 2019–2020 campaigns, we estimate that the immunization efforts prevented 72,000 (95% CI 58,000 to 86,000) case reports from 2019 to 2023. Leveraging our estimate of the reporting rate from the previous section, this corresponds to 1.3 million expected infections, and with the same conservative estimate as above for the infection–fatality ratio for measles [15], this implies that the 2019–2020 campaigns prevented over 10,000 deaths in Somalia.

## 5. Conclusions

The measles vaccine already accounts for over 80% of the deaths averted among vaccine-preventable diseases [18], and the WHO’s SAGE has stressed the importance of timely, high-coverage measles campaigns to achieve high global population immunity against infection [1]. Recent downward trends in routine vaccination coverage and increases in measles burden emphasize the current threats to measles elimination goals. This is a critical time to understand and improve vaccine delivery methods in the world’s most challenging settings.

This study makes it clear that global partnerships with national governments and local implementing agencies can deliver vaccines and save lives in the face of staggering complexity. The protracted nature of Somalia’s crises has led to a fragile health system with fragmented immunization. The planning, organization, and implementation of successful campaigns against measles in this country in particular, reaching children living in inaccessible and security-compromised areas, is a show of exceptional resilience, strength, and determination.

## Figures and Tables

**Figure 1 vaccines-12-00314-f001:**
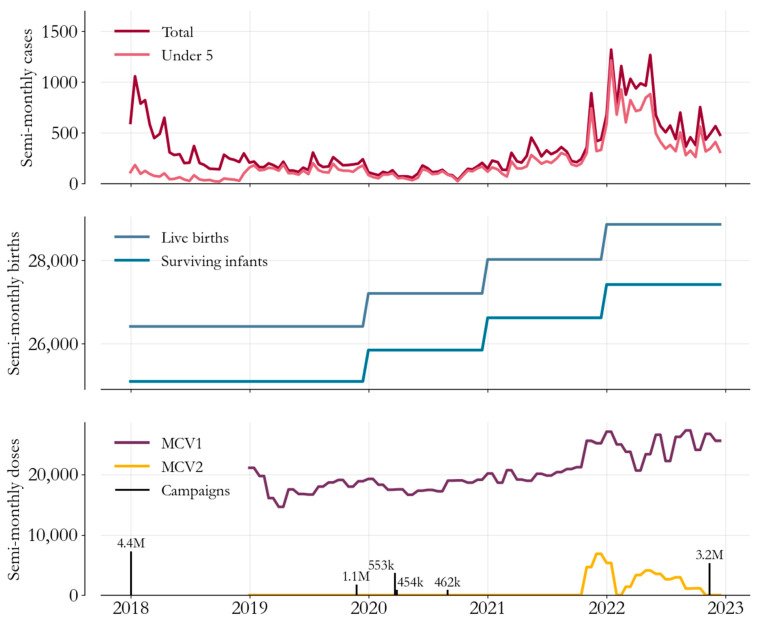
Measles cases, births, and vaccinations, Somalia, 2018–2023. Top panel: Case-based surveillance data (red) illustrate the concentration of burden in children under 5 (pink). Middle panel: estimates of birth rates show a recent growth in birth cohort size. Bottom panel: routine first (purple) and second (yellow) dose delivery has been augmented by vaccine campaigns (black bars).

**Figure 2 vaccines-12-00314-f002:**
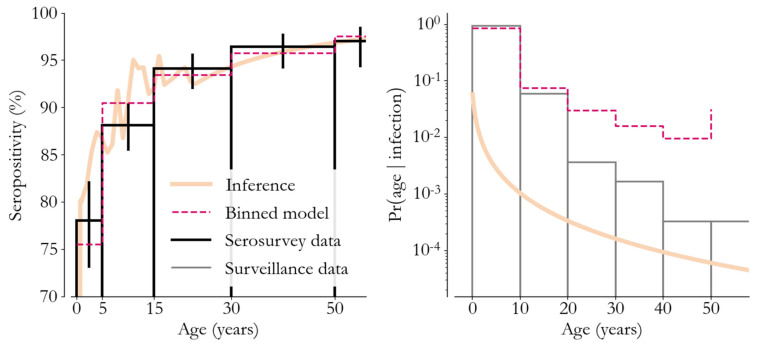
Measles serological survey, Somalia, 2021. Serological data (black) show that 95% of Somalians have immunity against measles before turning 30. This data give us insight into the balance between vaccination- and infection-derived immunity.

**Figure 3 vaccines-12-00314-f003:**
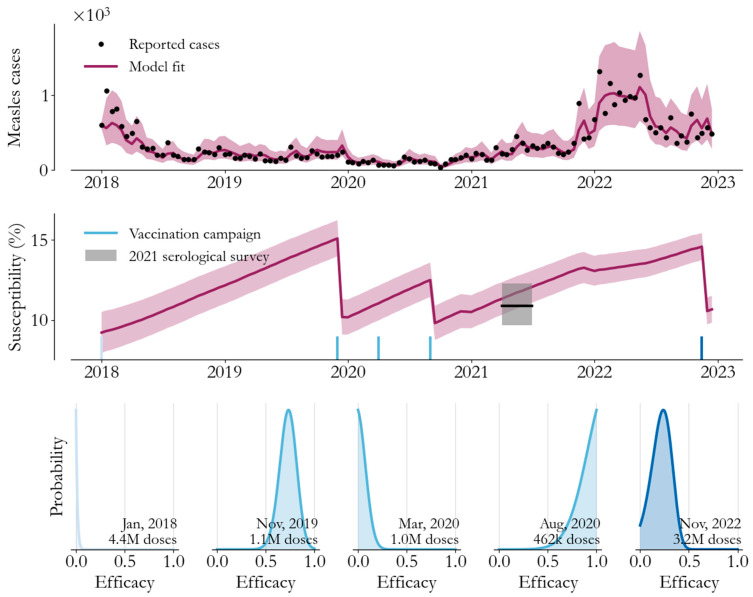
Modelling measles transmission, Somalia, 2018–2023. Top panel: transmission model samples (mean red line, 95% confidence interval shaded) closely follow observed cases (black dots). Middle panel: the fraction of the population susceptible over time incorporates vaccination campaigns (blue) and agrees with the serological survey results (black line, 95% confidence interval shaded). Bottom panel: the fraction of campaign doses that immunized a susceptible child varies across campaigns (blue shading groups campaign efforts).

**Figure 4 vaccines-12-00314-f004:**
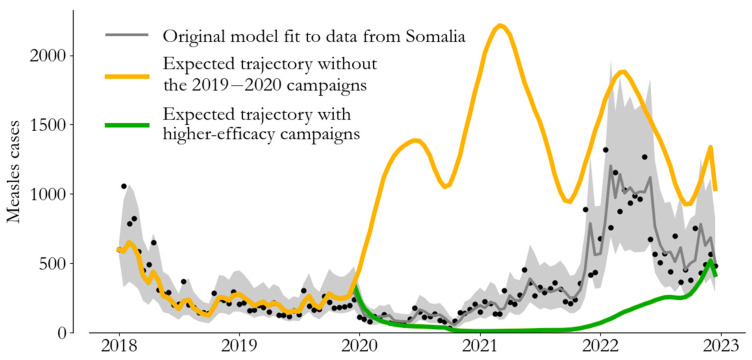
Model-based counter-factual epidemiology. We can compare the fitted model (grey) to alternative scenarios without the 2019–2020 campaigns (yellow) and with 50% more effective 2019–2020 campaigns (green) to estimate the impact in terms of averted burden.

## Data Availability

The data presented in this study are available on request from the corresponding author (N.T.), given permission from the Federal Ministry of Health, Somalia. The data are not publicly available for privacy reasons.

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
