# Peer review of "Estimating the Impact of Vaccination Campaigns on Measles Transmission in Somalia"

_vaccines, 2024, doi:10.3390/vaccines12030314_

Round 1
Reviewer 1 Report
Comments and Suggestions for Authors
In this manuscript, the authors assess the impact of campaign interventions on Somalia’s measles susceptibility, explore a counter-factual epidemiology without the 2019—2020 campaigns and estimate those interventions prevented over 10000 deaths using Somalia-specific model of measles transmission. This paper clearly showed that global partnerships with national governments and local implementing agencies can deliver vaccines and save lives in the face of staggering complexity.
Figure 2, Y axis does not start from 0. Also, please label the age according to the column.
Author Response
Thank you for your review of our manuscript.
As far as your recommendations, we've modified Figure 2's x-axis to better reflect the bars. While it is true that the y-axis does not start from 0, doing so makes distinction between the model and the data less visible, and we overall feel the current y-axis is clearer. We have left that portion of the figure as is.
Reviewer 2 Report
Comments and Suggestions for Authors
This short communication by Thakkar and colleagues report on a modeling approach to estimate the impact of measles vaccination campaigns in Somalia. The measles transmission model combines case-based surveillance, serosurveillance data, and the vaccine registry. The model successfully quantifies the estimated impact of the measles campaigns conducted in Somalia from 2018-2023, demonstrating that 1 in every 5 vaccine doses immunized a susceptible child. Using the impact data, the authors also model the disease trajectory in the absence of campaigns, illustrating that > 10,000 measles-related deaths were prevented. This is an important commentary to add to the literature, as an example of measles vaccine campaign impact in countries with fragile health systems. I approve of acceptance for publication after the authors address minor revisions listed below:
Lines 4-20: Why is the author line formatted this way? It’s standard practice to list all names with a superscript number that can be linked to institutional affiliation. The different affiliations should then be listed under the author line in numerical order.
Line 36-37: Minor but the wording of this sentence indicates that all suspected measles cases are true cases. Suggest a slight re-wording to indicate that measles cases are preventable through vaccination but making it clear that all suspect cases are not true measles cases.
Lines 74-92: Rather than referring to Figure 1 panels as if you are walking through them in a presentation, just state the information presented in the Figure. For example: “We see that growth cohorts have grown in size since 2020…” should just be “Growth cohorts have grown in size since 2020…”. You can reference which panel by just including in parentheses when discussing relevant information (e.g., (Fig. 1, Panel 3).
Figure 2 Caption: This needs to include more detail to explain the data in the figure. The text in the main body of the Communication does provide context, but the figure caption should be detailed enough to stand alone from the text and provide the reader with enough information to interpret the data.
Lines 130-132: The approaches from these references should ideally be briefly summarized in the text here with a short phrase (e.g., the name of the approach, type of model used, etc.) rather than just saying “the approach used in Ref 10”.
Line 158: Should say susceptible child?
Figure 3: Why is the Nov 2022 curve a different shade of blue than the other figures?
Figure 4 Caption: Minor. Font sizing is inconsistent.
Line 195: Rephrase to “immunity against infection”
Author Response
Thank you for your thorough review and comments.
Responding point-by-point:
1. Lines 4-20: We are not sure what the appropriate author format is, this format was chosen by the journal, and we defer to the editor on this point. We can easily reformat given a recommendation.
2. Lines 36-37: This is a very important and clarifying point. We've edited these lines appropriately.
3. Lines 74-92: While we appreciate that some papers are written in the passive voice suggest by the reviewer, we think this is a stylistic choice, and other reviewers have complemented the writing specifically (see Reviewer 3). We have left the presentation-like style as is.
4. Figure 2 Caption: The captions throughout the manuscript are sparse relative to the text, which was a choice we made to fit the length requirements of the Short Communication format. While we appreciate that many papers try to have captions detailed enough for stand-alone reading, we again feel this is a stylistic choice, and we have left the caption as is.
5. Lines 130-132: Those references don't provide names for the bespoke approaches used to fit the model. Rather than introduce a name, we feel it's more accurate and clearer to cite the relevant papers.
6. Line 158: Thank you - we have added this edit.
7. Figure 3: We were attempting to group simultaneously planned campaign efforts by shade of blue. We've decreased the contrast in the figure, and we've added clarifying text to the caption.
8. Figure 4 Caption: Thank you - we've fixed this issue.
9. Line 195: We've incorporated this edit.
Reviewer 3 Report
Comments and Suggestions for Authors
The manuscript is exceptionally well written and merits prompt publication. My only comment is that the authors seem to shift between referring to vaccination and immunisation in the title and text of the manuscript. I know that the SAGE prefers to use the term “immunisation” when discussing public health interventions to protect individuals from infection, but the choice of term is really up to the authors, as long as it is consistent.
Author Response
Thank you for your thoughts on our paper.
We appreciate the feedback to be intentional about our word choice on immunization and vaccination. Looking over the manuscript, we've tried to consistently use "vaccination" when we are referring to an intervention being delivered, and we use "immunization" in the context of a susceptible child being seroconverted. Reading the paper with this in mind, we edited line 71 and left the rest as is.